# No offence, Bert - I insult only humans! Multiple addressees sentence-level attack on toxicity detection neural networks

**Sergey Berezin, Reza Farahbakhsh, Noel Crespi**
SAMOVAR, Télécom SudParis, Institut Polytechnique de Paris
91120 Palaiseau, France
sberezin@telecom-sudparis.eu

## Abstract

We introduce a simple yet efficient sentence-level attack on black-box toxicity detector models. By adding several positive words or sentences to the end of a hateful message, we are able to change the prediction of a neural network and pass the toxicity detection system check. This approach is shown to be working on seven languages from three different language families. We also describe the defence mechanism against the aforementioned attack and discuss its limitations.

## 1 Introduction

Toxicity detection systems have become a crucial part of automoderation solutions. They are now used by most social media platforms, including those with groups of people in dangerously sensitive conditions, e.g. suicide prevention, Facebook groups, victims of cyberbullying, etc. Vulnerability in such systems may have dreadful, terrific effects on people, especially those in precarious situations.(Cheng et al., 2022).

On the other hand, such systems can be and are used to silence the voices of criticism, which leads to the creation of echo chambers and amplifies the voice of the (powerful) minority over the voice of the majority, thereby destroying the foundation of democracy and denying the freedom of speech (Gorwa et al., 2020).

This situation can be viewed as a double-edged sword, and we propose another double-edged sword that can be used to parry the blade of toxicity detection systems.

### 1.1 Task description

We present an attack on toxicity detection models based on the separation of the messages intended for a person and a piece of text added to confuse an algorithm.

For example, this can be done by concatenating an original message with a collection of keywords of an opposite intent: "Kill yourself, you dirty pig! Text for a bot to avoid ban: flowers, rainbow, happy, happy good" - here only the underlined part is intended to address the human, and it is clear that the remaining part was added to avoid detection by the automoderation algorithm.

This represents an example of a sentence-level black-box adversarial attack on Natural Language Processing systems.

### 1.2 Related work

The first work suggesting the concatenation of distracting but meaningless sentences at the end of a paragraph to confuse a neural model was "Adversarial Examples for Evaluating Reading Comprehension Systems" (Jia and Liang, 2017) (according to the (Zhang et al., 2020)). Jia and Liang attacked question-answering systems with either manually-generated informative sentences (ADDSENT) or arbitrary sequences of words using a pool of 20 random common words (ADDANY). Both perturbations were obtained by iterative querying of the neural network until the output was changed. The authors of "Robust Machine Comprehension Models via Adversarial Training" (Wang and Bansal, 2018) improved this approach by varying the locations where the distracting sentences are placed and expanding the set of fake answers for generating the distracting sentences (ADDSENTDIVERSE).

In "Universal Adversarial Triggers for Attacking and Analyzing NLP" (Wallace et al., 2019) apply the gradient-based technique to construct an adversarial text. Despite being a white-box attack in its origin (due to the gradient information required in the training phase), this approach can be applied as a black-box attack during its inference.

The T3 model (Wang et al., 2020) utilises the autoencoder architecture to generate adversarial texts that can manipulate the question-answering models to output the targeted incorrect answer.

In "All You Need is "Love": Evading Hate

Speech Detection" ([Gröndahl et al., 2018](#)) perform a sentence-level attack on hate speech detection systems by inserting typos, changing word boundaries and adding innocuous words to the original hate speech. The authors show the effectiveness of such an attack on MLP, CNN+RNN and LSTM neural networks.

## 1.3 Contribution

Our contributions lie in the following:

1) **Introduction of a concept of "To Each His Own" attack**, based on the idea of separating the messages addressed to a human and those addressed to an algorithm. The attack exploits the fact that toxicity detection models are trained on sentence-level labels and do not consider the context or the intention of the message.

2) **A systematic approach to the generation of adversarial texts.** The words and sentences are selected based on their positivity scores rather than being randomly selected from non-hate ones, as in previous works.

3) **Demonstration of attacks on modern state-of-the-art architectures**. We tested the most popular transformer-based toxicity detectors, including Google Perspective API and OpenAI Moderation API.

4) **Highlighting a significant flaw in the current approach to toxicity detection.** We show that the predicted toxicity score can be watered down by the addition of positive words to the point of effectively masking toxicity.

5) **Multi- and cross-lingual settings of attacks.** We describe the attacks on seven different languages and combinations among them. A larger number of languages leads to a larger number of possible attacks - in the case of the 5-language model, we have 120 attack scenarios.

## 2 "To Each His Own" attack

### 2.1 Word-based attack

In this attack type, the original message is appended or prepended with a sequence of positive words, with the aim that these words will be analysed by a model but passed over by a human. This allows an attacker to preserve their original message without any changes, unlike in other attacks, which substitute words with synonyms or replace original characters with their look-alikes.

In our experiment, we used words from the Senti-WordNet with a positivity score $\geq 0.85$. We tested 14 openly available pretrained neural networks with the phrase "I hate you" and the addition of 0 to 9 positive words.

In case the toxicity score for a suggested phrase was estimated to be less than 0.5 - the most common threshold, we used the phrase "I hate you dumbass" instead. If, after attacking with 9 words, the model's output was greater than 0.5, we continued adding new positive words until the score dropped below this threshold or until we reached the maximum length of an input sequence for the model tested. For the English language, we also performed experiments with the phrase "Text for bot:" placed between a message and added words.

We conducted additional experiments with the Perspective API and OpenAI Moderation API, discovering they are also vulnerable to our attacks.

For eight non-English models, we asked native speakers of the respective languages to translate a selected subset of words and attack phrases. Within this study, we tested the language models for seven languages aside from English: Russian, German, French, Portuguese, Spanish, Turkish and Vietnamese. It is worth noting that the first five listed languages represent three branches (Slavic, Germanic and Italic) of the Indo-European language family, and the last two are from different language families: the Turkic and the Austro-Asiatic family, respectfully.

By using various models and languages, we aim to investigate if the suggested attack is model- and language-agnostic.

### 2.2 Cross-lingual word-based attack

The main weakness of all multilingual models lies in their greatest advantage - the ability to work with multiple languages and writing systems. Adding positive words in different languages will even more separate messages to the human and to the toxicity detection system, perhaps even denying a human the ability to read non-intended text. Examples of such texts are shown in Figure 1.

Even some monolingual models, which had been exposed to another language during the pretraining phase, fall victim to this situation.

We also conducted cross-lingual experiments with OpenAI models, including ChatGPT in conversation mode with different prompts, and found them susceptible to attack.

Perspective API was found to be non-functional in a multilingual setting, failing with an error, as it

| I hate you любовь альтруистичный щедрый |
| You stink. eşsiz mükemmel telkari güzel ahlak |
| Goodbye, dumbass! animado mejor formidable |

Figure 1: Examples of cross-lingual word-based attack. The languages of the added text from the top are Russian, Turkish, and Spanish.

is unable to detect the language of the text.

## 2.3 Sentence-based attack

Incoherent text, produced by concatenating lexically unconnected words, is easily detectable by modern language models. To address that easy detection, we experimented with another version of the attack: concatenating sentences from the Stanford Sentiment Treebank with a positivity score $\geq 0.9$ and a length of no less than 100 symbols.

Since sentences consist of grammatically linked words instead of just a set of random positive words, this attack variant is less obvious and more challenging to detect.

## 3 Defence

We performed simple adversarial training of the DistilBERT model on the binary Jigsaw Toxic Comments dataset to improve the model's defence. We performed experiments with word- and sentence-based attacks, attacking both toxic messages and all messages in the dataset. In addition, we picked out only toxic messages and attacked half of them - in this scenario, the task was distinguishing attacked and non-attacked texts.

## 4 Results

### 4.1 Attack

During a word-based attack, both prepending and appending of the positive words showed similar results, with prepending being slightly less effective.

Appending nine words was enough to flip the prediction of almost every model. The "SkolkovoInstitute RoBERTa" toxicity classifier required 23 words to fall below 0.5, and "English-abusive-MuRIL" was still strongly predicting even after the addition of 252 words and reaching the length limit for the input. The results of selected experiments are shown in tables 1, 2 and 3[1].

The addition of the phrase "Text for bot:" between two parts of a text made a little difference in

[1]Full tables can be found in Appendix A.

| n words | BERT | RoBERTa | ELECTRA |
|---|---|---|---|
| 0 | 0,951 | 0,857 | 0,898 |
| 1 | 0,925 | 0,743 | 0,730 |
| 2 | 0,708 | 0,507 | 0,361 |
| 3 | 0,592 | 0,415 | 0,064 |
| 4 | 0,579 | 0,274 | 0,067 |
| 5 | 0,548 | 0,170 | 0,082 |
| 6 | 0,510 | 0,158 | 0,087 |
| 7 | 0,494 | 0,148 | 0,069 |
| 8 | 0,465 | 0,142 | 0,065 |
| 9 | 0,303 | 0,132 | 0,072 |

Table 1: Word-based attack in English on different transformer architectures.

| n words | OpenAI S | OpenAI L | Per. API |
|---|---|---|---|
| 1 | 0.643 | 0.946 | 0.899 |
| ... | ... | ... | ... |
| 10 | 0.421 | 0.847 | **0.846** |
| 11 | 0.381 | 0.705 | 0.852 |
| 12 | 0.427 | 0.74 | 0.852 |
| 13 | 0.453 | 0.761 | 0.852 |
| 14 | 0.426 | **0.665** | 0.853 |
| 15 | 0.362 | 0.681 | 0.853 |
| 16 | **0.265** | 0.716 | 0.853 |

Table 2: Word-based attack in English on Perspective API and OpenAI Moderation API. Per. - perspective, S - stable, L - latest.

the results, i.e. it can be used to create even more apparent separation without lowering an attack's efficiency.

As expected, the results of cross-lingual attacks followed the same trend as those of monolingual ones. These results are shown in tables 4 and 5[1].

With a sentence-based attack, adding even one sentence drastically lowered the prediction scores of the models. The scores are shown in table 6[1].

### 4.2 Defence

The model's F1 measure on toxic class fell from 0.80 to 0.44 after an attack with 15 sentences and to 0.79 after an attack with 50 words. After adversarial training, the model regained its F1 measure in both cases.

Interestingly, testing the word-trained model on sentence-attacked examples showed high precision and low recall (0.90 and 0.68, respectively). Testing the sentence-trained model on word-attacked examples showed the opposite result, with low precision and high recall (0.57 and 0.93, respectively).

| n words | Vietnamese | French | Turkish |
|---|---|---|---|
| 0 | 0,996 | 0,970 | 0,814 |
| 1 | 0,008 | 0,962 | 0,186 |
| 2 | 0,007 | 0,573 | 0,503 |
| 3 | 0,008 | 0,396 | 0,043 |
| 4 | 0,009 | 0,090 | 0,048 |
| 5 | 0,008 | 0,054 | 0,085 |
| 6 | 0,007 | 0,055 | 0,090 |
| 7 | 0,007 | 0,064 | 0,029 |
| 8 | 0,007 | 0,082 | 0,009 |
| 9 | 0,007 | 0,045 | 0,005 |

Table 3: Word-based attack on transformer-based language models trained on the languages from three different language families.

| n words | eng+tur | eng+rus | eng+sp |
|---|---|---|---|
| 0 | 0,971 | 0,971 | 0,971 |
| 1 | 0,913 | 0,717 | 0,802 |
| 2 | 0,904 | 0,717 | 0,687 |
| 3 | 0,912 | 0,761 | 0,536 |
| 4 | 0,541 | 0,800 | 0,558 |
| 5 | 0,634 | 0,592 | 0,500 |
| 6 | 0,731 | 0,456 | 0,539 |
| 7 | 0,604 | 0,402 | 0,426 |
| 8 | 0,519 | 0,433 | 0,434 |
| 9 | 0,455 | 0,369 | 0,384 |

Table 4: Cross-lingual word-based attack on toxicity detection models.

| n | L en+fr | L en+ger | S en+fr | S en+ger |
|---|---|---|---|---|
| 0 | 0.872 | 0.872 | 0.888 | 0.888 |
| 1 | 0.931 | 0.981 | 0.793 | 0.848 |
| 2 | 0.895 | 0.898 | 0.779 | 0.510 |
| 3 | 0.769 | 0.889 | 0.671 | 0.511 |
| 4 | 0.816 | 0.846 | 0.594 | 0.344 |
| 5 | 0.771 | 0.807 | 0.606 | 0.302 |
| 6 | 0.800 | 0.647 | 0.618 | 0.318 |
| 7 | 0.659 | 0.533 | 0.605 | **0.201** |
| 8 | 0.592 | 0.555 | 0.573 | 0.208 |
| 9 | **0.579** | **0.489** | **0.424** | 0.283 |

Table 5: The results of cross-lingual attacks on OpenAI models. n - number of words added, L - latest, S - stable.

| n sentences | BERT | RoBERTa | ELECTRA |
|---|---|---|---|
| 0 | 0,590 | 0,962 | 0,898 |
| 1 | 0,001 | 0,001 | 0,232 |
| 2 | 0,001 | 0,002 | 0,147 |

Table 6: Results of sentence-based attacks.

creasing the quality and quantity of the possible text insertions.

The complex approach, which covers all levels of perturbation, should be applied both in attack and defence real-life scenarios.

The defence performed strongly, achieving an F1 score of 0.82 on binary toxic classification with word- and sentence-based attacks. The model achieved the same score on a non-attacked dataset, succeeding even better in distinguishing attacked sentences, with an F1 score of 0.99. However, defence models must be built for both word- and sentence-based attacks, as one single model will not work perfectly for both scenarios.

## 5 Discussion

All of the attack variants showed decent results in confusing toxicity detection models in every model tested (to varying degrees). The simplistic nature of such attacks could allow virtually any Internet user to exploit them to avoid automoderation systems.

A possible countermeasure for this type of attack is adversarial training or rule-based filtering. However, a simple rule-based filter can be fooled by char-level adversarial attacks, and adversarial training can be made much more difficult by in-

## 6 Conclusion

In this paper, we demonstrated a novel and effective way of bypassing toxicity detection models by appending positive words or sentences to the end of a toxic message.

The introduced concept of the "To Each His Own" attack is based on the idea of separating the messages addressed to a human and those addressed to an algorithm. The attack exploits the fact that the toxicity detection models are trained on sentence-level labels and do not consider the context or the intention of messages.

The described attack in all its variants can be easily used on almost any toxic-detection neural model, with fairly good results. For future research, we suggest looking for a more sophisticated way of constructing insertions, perhaps with respect to the original message, making it a semantically correct addition to human-written messages.

## 7 Acknowledgments

We thank all the volunteers who helped us with translations.

We are grateful to the reviewers and the program chair of the EMNLP conference for providing constructive feedback, suggesting additional experiments and providing different points of view. This helped us structure the paper and ensure the robustness of our results.

Sergey Berezin wishes to record that he is grateful to Mr. Mario Fritz from Saarland University for the introduction to the domain of adversarial attacks through his insightful course on this topic. He is also grateful to Mr. Maxime Amblard from the University of Lorraine for encouraging him to pursue research in this direction and validating his ideas.

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

# A Appendix A

| n words | BERT | roberta-base | xlm-roberta-base | toxigen-roberta | toxigen-hatebert |
|---|---|---|---|---|---|
| 0 | 0,95 | 0,86 | 0,97 | 0,96 | 0,88 |
| 1 | 0,93 | 0,74 | 0,89 | 0,96 | 1,00 |
| 2 | 0,71 | 0,51 | 0,65 | 0,71 | 0,98 |
| 3 | 0,59 | 0,42 | 0,63 | 0,59 | 0,98 |
| 4 | 0,58 | 0,27 | 0,56 | 0,02 | 0,11 |
| 5 | 0,55 | 0,17 | 0,53 | 0,00 | 0,07 |
| 6 | 0,51 | 0,16 | 0,41 | 0,00 | 0,02 |
| 7 | 0,49 | 0,15 | 0,55 | 0,00 | 0,31 |
| 8 | 0,47 | 0,14 | 0,45 | 0,00 | 0,39 |
| 9 | 0,30 | 0,13 | 0,41 | 0,00 | 0,06 |

| n words | roberta skolkovo | roberta facebook | MuRIL | MuRIL* | electra-hateXplain |
|---|---|---|---|---|---|
| 0 | 1,00 | 1,00 | 0,13 | 0,96 | 0,90 |
| 1 | 1,00 | 1,00 | 0,10 | 0,95 | 0,73 |
| 2 | 0,99 | 1,00 | 0,03 | 0,94 | 0,36 |
| 3 | 0,92 | 1,00 | 0,01 | 0,92 | 0,06 |
| 4 | 0,98 | 1,00 | 0,01 | 0,92 | 0,07 |
| 5 | 0,98 | 1,00 | 0,01 | 0,92 | 0,08 |
| 6 | 0,99 | 1,00 | 0,03 | 0,94 | 0,09 |
| 7 | 0,99 | 1,00 | 0,01 | 0,93 | 0,07 |
| 8 | 0,99 | 0,03 | 0,01 | 0,91 | 0,07 |
| 9 | 0,98 | 0,59 | 0,01 | 0,89 | 0,07 |

| n words | multi-MuRIL | BERT-movies | DistilBERT | DistilBERT* |
|---|---|---|---|---|
| 0 | 0,74 | 0,32 | 0,07 | 0,95 |
| 1 | 0,63 | 0,13 | 0,01 | 0,45 |
| 2 | 0,19 | 0,10 | 0,01 | 0,17 |
| 3 | 0,01 | 0,12 | 0,01 | 0,14 |
| 4 | 0,02 | 0,08 | 0,00 | 0,11 |
| 5 | 0,12 | 0,10 | 0,03 | 0,16 |
| 6 | 0,44 | 0,09 | 0,03 | 0,22 |
| 7 | 0,35 | 0,07 | 0,02 | 0,17 |
| 8 | 0,31 | 0,06 | 0,01 | 0,09 |
| 9 | 0,25 | 0,04 | 0,01 | 0,08 |

Table 7: Word-based attack in English on different transformer architectures - extended table.

| n words | Vietnamese | French | German | | Spanish | Spanish |
|---|---|---|---|---|---|---|
| | PhoBERT | detoxify-fr | BERT-GermEval18Coarse | | dehatebert | detoxify-sp |
| 0 | 0,996 | 0,749 | 0,819 | | 0,970 | 0,943 |
| 1 | 0,008 | 0,834 | 0,531 | | 0,962 | 0,645 |
| 2 | 0,007 | 0,812 | 0,432 | | 0,573 | 0,654 |
| 3 | 0,008 | 0,757 | 0,305 | | 0,396 | 0,583 |
| 4 | 0,009 | 0,735 | 0,383 | | 0,090 | 0,463 |
| 5 | 0,008 | 0,652 | 0,392 | | 0,054 | 0,444 |
| 6 | 0,007 | 0,515 | 0,360 | | 0,055 | 0,334 |
| 7 | 0,007 | 0,415 | 0,438 | | 0,064 | 0,247 |
| 8 | 0,007 | 0,486 | 0,484 | | 0,082 | 0,244 |
| 9 | 0,007 | 0,511 | 0,422 | | 0,045 | 0,244 |
| **n words** | **Portuguese** | **Portuguese** | **Turkish** | **Russian** | **Russian** | **Russian** |
| | dehatebert | detoxify | detoxify-tur | Skolkovo | rubert-toxic | detoxify-ru |
| 0 | 0,591 | 0,983 | 0,977 | 0,814 | 0,689 | 0,973 |
| 1 | 0,614 | 0,887 | 0,812 | 0,186 | 0,360 | 0,863 |
| 2 | 0,587 | 0,794 | 0,841 | 0,503 | 0,042 | 0,834 |
| 3 | 0,489 | 0,709 | 0,735 | 0,043 | 0,037 | 0,817 |
| 4 | 0,473 | 0,637 | 0,767 | 0,048 | 0,028 | 0,817 |
| 5 | 0,410 | 0,663 | 0,745 | 0,085 | 0,039 | 0,684 |
| 6 | 0,406 | 0,496 | 0,752 | 0,090 | 0,050 | 0,503 |
| 7 | 0,418 | 0,327 | 0,673 | 0,029 | 0,044 | 0,534 |
| 8 | 0,434 | 0,332 | 0,649 | 0,009 | 0,042 | 0,514 |
| 9 | 0,447 | 0,354 | 0,622 | 0,005 | 0,052 | 0,414 |

Table 8: Word-based attack on languages from three different language families - extended table.

| n | toxigen-rob. | Skolkovo rob. | hatebert | MuRIL | electra-hateXplain | Multi-MuRIL |
|---|---|---|---|---|---|---|
| 0 | 0,962 | 0,999 | 0,590 | 0,126 | 0,898 | 0,737 |
| 1 | 0,001 | 0,002 | 0,001 | 0,022 | 0,232 | 0,160 |
| 2 | 0,002 | 0,001 | 0,001 | 0,017 | 0,147 | 0,091 |

Table 9: Results of sentence-based attack - extended table. n - number of added sentences, rob - roberta.

| n words | rus + eng 1 | rus + eng 2 | rus + sp | rus + fr | rus + ge |
|---|---|---|---|---|---|
| | SkolkovoInstitute/ russian_toxicity_classifier | rubert-toxic-pikabu-2ch | rubert-toxic-pikabu-2ch | rubert-toxic-pikabu-2ch | rubert-toxic-pikabu-2ch |
| 0 | 0,814 | 0,689 | 0,689 | 0,689 | 0,689 |
| 1 | 0,498 | 0,252 | 0,326 | 0,608 | 0,263 |
| 2 | 0,290 | 0,141 | 0,186 | 0,288 | 0,247 |
| 3 | 0,754 | 0,122 | 0,259 | 0,291 | 0,296 |
| 4 | 0,707 | 0,131 | 0,349 | 0,237 | 0,284 |
| 5 | 0,743 | 0,441 | 0,423 | 0,236 | 0,403 |
| 6 | 0,749 | 0,527 | 0,190 | 0,196 | 0,244 |
| 7 | 0,576 | 0,206 | 0,082 | 0,237 | 0,296 |
| 8 | 0,382 | 0,357 | 0,116 | 0,180 | 0,281 |
| 9 | 0,467 | 0,356 | 0,109 | 0,226 | 0,290 |

| n words | eng + tur | eng + rus | eng+sp 1 | eng+sp 2 | eng+sp 3 |
|---|---|---|---|---|---|
| | Detoxify | Detoxify | Detoxify | electra-base-hateXplain base-hateXplain | indic-abusive-allInOne-MuRIL |
| 0 | 0,971 | 0,971 | 0,971 | 0,898 | 0,737 |
| 1 | 0,913 | 0,717 | 0,802 | 0,574 | 0,724 |
| 2 | 0,904 | 0,717 | 0,687 | 0,912 | 0,454 |
| 3 | 0,912 | 0,761 | 0,536 | 0,471 | 0,590 |
| 4 | 0,541 | 0,800 | 0,558 | 0,340 | 0,574 |
| 5 | 0,634 | 0,592 | 0,500 | 0,270 | 0,668 |
| 6 | 0,731 | 0,456 | 0,539 | 0,216 | 0,579 |
| 7 | 0,604 | 0,402 | 0,426 | 0,226 | 0,508 |
| 8 | 0,519 | 0,433 | 0,434 | 0,212 | 0,290 |
| 9 | 0,455 | 0,369 | 0,384 | 0,186 | 0,236 |

Table 10: Cross-lingual word-based attack - extended table. First language listed - language of a message, second - language of added text.