# OpenReview forum: "No offence, Bert - I insult only humans! Multilingual sentence-level attack on toxicity detection networks"
_EMNLP/2023/Conference — EMNLP 2023 Findings_

### Official Review · Reviewer_KyeZ · 2023-08-05

**Soundness:** 3

**Excitement:**

2: Mediocre: This paper makes marginal contributions (vs non-contemporaneous work), so I would rather not see it in the conference.

**Paper Topic And Main Contributions:**

This paper presents an innovative yet straightforward method of attacking toxicity detector models by appending positive words or phrases to hateful messages. From the reported experiments, this approach could significantly challenge the effectiveness of neural network-based toxicity detectors.

**Questions For The Authors:**

Refer to the reasons to reject above.

**Reasons To Accept:**

1. The proposed method, though straightforward, appears to have a potent attacking impact.
2. The authors have also considered the implications of their method on multilingual models.

**Reasons To Reject:**

1. The toxicity detectors utilized in the study are self-constructed by the authors and appear to lack robustness. This factor potentially weakens the study's conclusions. It would be more compelling if the authors tested more popular and robust toxicity detectors, such as the Perspective API and other classifiers referenced in their related work.
2. While the authors' observation that the toxicity score decreases as the number of appended positive words increases seems intuitive, it may not necessarily be a novel insight. It is expected that the ratio of toxic words to the total number of words in a sentence would impact the toxicity score. The question then becomes whether this finding offers any new understanding of how toxicity detectors function or can be manipulated. Without further exploration or explanation, this observation might be considered an expected outcome rather than a groundbreaking discovery.
3. While the authors emphasize "multilingual" aspects in their paper, the reported results do not indicate that multilingual models are more susceptible than monolingual models. Furthermore, the proposed method doesn't seem to have a significant correlation with the multilingual setup.
4. The defense strategy employed (training on adversarial sampling) is fairly standard, and its effectiveness has been well-documented in previous research. Thus, it does not add a novel contribution to the field.



**Reproducibility:**

4: Could mostly reproduce the results, but there may be some variation because of sample variance or minor variations in their interpretation of the protocol or method.

**Reviewer Confidence:**

4: Quite sure. I tried to check the important points carefully. It's unlikely, though conceivable, that I missed something that should affect my ratings.

---

> ### Author Rebuttal · Authors · 2023-08-28
>
> Thank you for your review!
>
> We will answer your questions and considerations one by one.
>
> &nbsp;
> ## Reason to reject №1
> &nbsp;
> >1. The toxicity detectors utilized in the study are self-constructed by the authors and appear to lack robustness. This factor potentially weakens the study's conclusions. It would be more compelling if the authors tested more popular and robust toxicity detectors, such as the Perspective API and other classifiers referenced in their related work.
>
> We used the most popular toxicity detectors from the Hugging Face hub for each language tested and provided the names of these models. Also, we conducted additional experiments with the Perspective API and OpenAI Moderation API. As we discovered, they are also vulnerable to our attacks:
>
> In the table below we show the performance of a word-adding attack with the phrase: "I hate you, dumbass! text for a bot to avoid ban:"
>
> | n words | OpenAI stable | OpenAI latest | Perspective API |
> |:-------:|:-------------:|:-------------:|:---------------:|
> |    1    |     0.643     |     0.946     |      0.899      |
> |    2    |     0.663     |     0.915     |      0.903      |
> |    3    |     0.647     |     0.918     |      0.899      |
> |    4    |     0.605     |     0.904     |      0.886      |
> |    5    |     0.601     |     0.872     |      0.878      |
> |    6    |     0.553     |     0.902     |       0.87      |
> |    7    |     0.553     |     0.926     |      0.863      |
> |    8    |     0.551     |     0.916     |      0.854      |
> |    9    |      0.33     |     0.854     |      0.853      |
> |    10   |     0.421     |     0.847     |      **0.846**      |
> |    11   |     0.381     |     0.705     |      0.852      |
> |    12   |     0.427     |      0.74     |      0.852      |
> |    13   |     0.453     |     0.761     |      0.852      |
> |    14   |     0.426     |     **0.665**     |      0.853      |
> |    15   |     0.362     |     0.681     |      0.853      |
> |    16   |    **0.265**    |     0.716     |      0.853      |
>
> We also conducted additional experiments in other languages and between them (only with OpenAI, Perspective API is failing with an error in the multilingual setting). In the table below, you can find the results of cross-lingual attacks on OpenAI models:
>
> | n words | Latest EN+FR | Latest EN+GER | Stable EN+FR | Stable EN+GER |
> |:-------:|:------------:|:-------------:|:------------:|---------------|
> |    0    |     0.872    |     0.872     |     0.888    | 0.888         |
> |    1    |     0.931    |     0.981     |     0.793    | 0.848         |
> |    2    |     0.895    |     0.898     |     0.779    | 0.51          |
> |    3    |     0.769    |     0.889     |     0.671    | 0.511         |
> |    4    |     0.816    |     0.846     |     0.594    | 0.344         |
> |    5    |     0.771    |     0.807     |     0.606    | 0.302         |
> |    6    |      0.8     |     0.647     |     0.618    | 0.318         |
> |    7    |     0.659    |     0.533     |     0.605    | **0.201**         |
> |    8    |     0.592    |     0.555     |     0.573    | 0.208         |
> |    9    |     **0.579**    |    **0.489**     |     **0.424**    | 0.283         |
>
> We also tested ChatGPT in conversation mode with different prompts and found it susceptible to the attack.
>
> &nbsp;
> ## Reason to reject №2
> &nbsp;
>
> >While the authors' observation that the toxicity score decreases as the number of appended positive words increases seems intuitive, it may not necessarily be a novel insight. It is expected that the ratio of toxic words to the total number of words in a sentence would impact the toxicity score. The question then becomes whether this finding offers any new understanding of how toxicity detectors function or can be manipulated. Without further exploration or explanation, this observation might be considered an expected outcome rather than a groundbreaking discovery.
>
> Our intuition, built on understanding the principles of current text classification models, indeed tells us that the total number of non-toxic words in a sentence would impact the toxicity score.\
> &nbsp;
>
> However, we believe it is not how it should be:\
> &nbsp;
>
> Jigsaw defines toxicity as "rude, disrespectful, or unreasonable language that is likely to make someone leave a discussion." - From this point of view, messages: "Nobody wants to talk with you!" and "Nobody wants to talk with you! text for a bot to avoid ban: flowers happiness good love" should have the same toxicity score.\
> &nbsp;
>
> In the same way as a drop of poison will turn a bottle of wine into a bottle of poison, a toxic message can not be watered down by a bunch of positive words. Hatred cannot be justified.
>
> Thus, by our attack we demonstrate a significant flaw in the current approach to toxicity detection.
>
> &nbsp;
> ## Reason to reject №3
> &nbsp;
>
> >While the authors emphasize "multilingual" aspects in their paper, the reported results do not indicate that multilingual models are more susceptible than monolingual models. Furthermore, the proposed method doesn't seem to have a significant correlation with the multilingual setup.
>
> Multilinguality of the model allows us to perform cross-lingual attacks, e.g. combining English offensive messages with positive words in Arabic. In this case, a larger amount of languages leads to a larger number of possible attacks - in the case of the 5-language model, we can have a factorial of 5 combinations of languages, making it 120 possible attack scenarios.
>
> Such a cross-lingual attack might even effectively prevent humans from reading the part which is intended to distract the model (if they cannot read Arabic), thus emphasizing the idea of clear separation of messages addressed to a human and an algorithm even more.
>
> &nbsp;
> ## Reason to reject №4
> &nbsp;
>
> >The defense strategy employed (training on adversarial sampling) is fairly standard, and its effectiveness has been well-documented in previous research. Thus, it does not add a novel contribution to the field.
>
> We wanted to show the applicability of this defence and to demonstrate that one needs to build defence models for both word- and sentence-based attacks, as one single model will not work perfectly for both scenarios.
>
> &nbsp;
>
> Thank you for your time and effort,\
> We rest at your disposal to answer any additional questions should you have any.

---

### Official Review · Reviewer_fLb3 · 2023-08-05

**Soundness:** 4

**Excitement:**

3: Ambivalent: It has merits (e.g., it reports state-of-the-art results, the idea is nice), but there are key weaknesses (e.g., it describes incremental work), and it can significantly benefit from another round of revision. However, I won't object to accepting it if my co-reviewers champion it.

**Paper Topic And Main Contributions:**

This paper proposes a simple adversarial attack for fooling toxicity detection models. The proposed method add positive sentiment words from Senti-WordNet to the end of toxic input until the target model flips predictions. The authors tested the method under both monolingual and cross-lingual settings. Experimental results show most models are fooled within 9 words. Further results on adversarial training show that the method can be easily defended.

**Questions For The Authors:**

1.	Line 179 to line 190, why adding SST sentences makes the added part more coherent?
2.	Line 240 to 245, for a toxic content detector, recall is more important than precision in many use cases, did you try changing the threshold to increase the recall?


**Reasons To Accept:**

1.	The paper is well-written, easy to follow and understand.
2.	The proposed method is simple yet effective, fooling most models within 9 words.
3.	Under cross-lingual settings, the proposed method has the potential of refusing the humans to read added words.


**Reasons To Reject:**

1.	Lack of novelty. Adding words to inputs to fool model has been proposed in many previous attacks.
2.	The proposed can be recognized by humans very easily. In real scenarios, defenders can find such attacks very soon and deploy a new filtering model very easily.


**Reproducibility:**

3: Could reproduce the results with some difficulty. The settings of parameters are underspecified or subjectively determined; the training/evaluation data are not widely available.

**Reviewer Confidence:**

4: Quite sure. I tried to check the important points carefully. It's unlikely, though conceivable, that I missed something that should affect my ratings.

---

> ### Author Rebuttal · Authors · 2023-08-28
>
> Thank you for your review!
>
> We will answer your questions and considerations one by one.
>
> &nbsp;
> ## Question №1
> &nbsp;
>
> >Line 179 to line 190, why adding SST sentences makes the added part more coherent?
>
> Because sentences consist of grammatically linked words instead of just a set of random positive words, as in a word-based attack scenario. This will make the attack less obvious.
>
> &nbsp;
> ## Question №2
> &nbsp;
>
> >Line 240 to 245, for a toxic content detector, recall is more important than precision in many use cases, did you try changing the threshold to increase the recall?
>
> The point of this paragraph is to show the applicability of this defence and to demonstrate that one needs to build defence models for both word- and sentence-based attacks, as one single model will not work perfectly for both scenarios. The threshold tuning will likely increase the recall in the price of the precision, but it is better to train two separate models.
>
> &nbsp;
> ## Reason to reject №1
> &nbsp;
>
> >1. Lack of novelty. Adding words to inputs to fool model has been proposed in many previous attacks.
>
> The key contributions lay in the following:
>
> 1) **A systematic approach to the generation of adversarial texts.**
> The words and sentences used were selected based on their positivity scores rather than being randomly selected from non-hate ones as in previous works.
>
>
> 2) **Multi- and cross-lingual setting of attack.**
> We described the attacks on 7 different languages and combinations between them. Multilinguality of the model allows us to perform cross-lingual attacks, e.g., combining English offensive messages with positive words in Arabic.\
> In this case, a larger number of languages leads to a larger number of possible attacks - in the case of the 5-language model, we can have 120 possible attack scenarios.
>
> 3) **Study of the dependency of models’ confidence on the length of an added sequence.**
> We show that the total number of non-toxic words in a sentence would impact the toxicity score.
> However, we believe it is not how it should be:\
> Jigsaw defines toxicity as "rude, disrespectful, or unreasonable language that is likely to make someone leave a discussion." from this point of view, messages: *"Nobody wants to talk with you!"* and *"Nobody wants to talk with you! text for a bot to avoid ban: flowers happiness good love"* should have the same toxicity score.\
> In the same way as a drop of poison will turn a bottle of wine into a bottle of poison, a toxic message cannot be watered down by a bunch of positive words. Hatred cannot be justified.\
> Thus, **by our attack we demonstrate a significant flaw in the current approach to toxicity detection**.
>
> &nbsp;
> ## Reason to reject №2
> &nbsp;
>
> >The proposed can be recognized by humans very easily. In real scenarios, defenders can find such attacks very soon and deploy a new filtering model very easily.
>
> This is the point of the suggested method: **our attack targets exclusively machines, making it clear to humans that the positive words are intended for the automoderation system** (e.g. "Nobody wants to talk with you! text for a bot to avoid ban: flowers happiness good love").
> This allows an attacker to preserve their original message without any changes, unlike in other attacks which substitute words with synonyms or replace original characters with their look-alike.
>
> Moreover, in a real-world scenario, an adversary can use this attack in combination with other approaches, thus increasing defence difficulty.
>
> In our current research, we are developing a more sophisticated way of constructing insertion with respect to the original message, making it a semantically correct addition to a human-written message, thus making adversarial training more difficult and increasing the variety of adversarial insertions. It’s a GAN-like architecture, with an LLM, such as Llama 2, acting as a generator and a toxicity detector acting as a discriminator.
>
>
>
> We conducted additional experiments with the Perspective API and OpenAI Moderation API. As we discovered, they are also vulnerable to our attacks.
> In the table below we show the performance of a word-adding attack with the phrase: "I hate you, dumbass! text for a bot to avoid ban:"
>
> | n words | OpenAI stable | OpenAI latest | Perspective API |
> |:-------:|:-------------:|:-------------:|:---------------:|
> |    1    |     0.643     |     0.946     |      0.899      |
> |    2    |     0.663     |     0.915     |      0.903      |
> |    3    |     0.647     |     0.918     |      0.899      |
> |    4    |     0.605     |     0.904     |      0.886      |
> |    5    |     0.601     |     0.872     |      0.878      |
> |    6    |     0.553     |     0.902     |       0.87      |
> |    7    |     0.553     |     0.926     |      0.863      |
> |    8    |     0.551     |     0.916     |      0.854      |
> |    9    |      0.33     |     0.854     |      0.853      |
> |    10   |     0.421     |     0.847     |      **0.846**      |
> |    11   |     0.381     |     0.705     |      0.852      |
> |    12   |     0.427     |      0.74     |      0.852      |
> |    13   |     0.453     |     0.761     |      0.852      |
> |    14   |     0.426     |     **0.665**     |      0.853      |
> |    15   |     0.362     |     0.681     |      0.853      |
> |    16   |    **0.265**    |     0.716     |      0.853      |
>
> We also conducted additional experiments in other languages and between them (only with OpenAI, Perspective API is failing with an error in the multilingual setting). In the table below, you can find the results of cross-lingual attacks on OpenAI models:
>
> | n words | Latest EN+FR | Latest EN+GER | Stable EN+FR | Stable EN+GER |
> |:-------:|:------------:|:-------------:|:------------:|---------------|
> |    0    |     0.872    |     0.872     |     0.888    | 0.888         |
> |    1    |     0.931    |     0.981     |     0.793    | 0.848         |
> |    2    |     0.895    |     0.898     |     0.779    | 0.51          |
> |    3    |     0.769    |     0.889     |     0.671    | 0.511         |
> |    4    |     0.816    |     0.846     |     0.594    | 0.344         |
> |    5    |     0.771    |     0.807     |     0.606    | 0.302         |
> |    6    |      0.8     |     0.647     |     0.618    | 0.318         |
> |    7    |     0.659    |     0.533     |     0.605    | **0.201**         |
> |    8    |     0.592    |     0.555     |     0.573    | 0.208         |
> |    9    |     **0.579**    |    **0.489**     |     **0.424**    | 0.283         |
>
> We also tested ChatGPT in conversation mode with different prompts and found it susceptible to the attack.
>
> &nbsp;
>
> Thank you for your time and effort,\
> We rest at your disposal to answer any additional questions should you have any.

---

### Official Review · Reviewer_RWV8 · 2023-08-06

**Soundness:** 2

**Excitement:**

3: Ambivalent: It has merits (e.g., it reports state-of-the-art results, the idea is nice), but there are key weaknesses (e.g., it describes incremental work), and it can significantly benefit from another round of revision. However, I won't object to accepting it if my co-reviewers champion it.

**Paper Topic And Main Contributions:**

This paper proposes a simple multilingual sentence-level attack on the black-box toxicity detector models. The attack is a simple way to add positive words or sentences to a hateful message. Experiments are conducted on 7 languages from 3 different language families.

**Questions For The Authors:**

1. How about the detectors used here in the paper? Maybe some more details need to be provided. Are they advanced enough?

2. How about the mentioned attacks for large language models? Will the detection be easy?

**Reasons To Accept:**

The problem of toxicity attack is interesting, and various toxicity attack models may need to be considered to improve the robustness of the detectors.

**Reasons To Reject:**

1. The contribution of the paper may be limited. Only a single positive words/sentences adding attack model is considered to improve the robustness of the toxicity detector. More attack models may be better.

2. The attack model may need to be justified a lot. One straightforward flaw is that the model can be easily identified by humans and not identified by machines. Some simple modifications of the detector may be able to solve such a problem; for example, a language model can be used first to check if the sentences sound natural, or natural language inference can be used first to detect if the relations between sentences are natural.

**Reproducibility:**

4: Could mostly reproduce the results, but there may be some variation because of sample variance or minor variations in their interpretation of the protocol or method.

**Reviewer Confidence:**

4: Quite sure. I tried to check the important points carefully. It's unlikely, though conceivable, that I missed something that should affect my ratings.

---

> ### Author Rebuttal · Authors · 2023-08-28
>
> Thank you for your review!
>
> We will answer your questions and considerations one by one.
>
> &nbsp;
> ## Reason to reject №1
> &nbsp;
>
> >1. The contribution of the paper may be limited. Only a single positive words/sentences adding attack model is considered to improve the robustness of the toxicity detector. More attack models may be better.
>
> The key contributions lay in the following:
>
> 1) **A systematic approach to the generation of adversarial texts.**
> The words and sentences used were selected based on their positivity scores rather than being randomly selected from non-hate ones as in previous works.
>
>
> 2) **Multi- and cross-lingual setting of attack.**
> We described the attacks on 7 different languages and combinations between them. Multilinguality of the model allows us to perform cross-lingual attacks, e.g., combining English offensive messages with positive words in Arabic.\
> In this case, a larger number of languages leads to a larger number of possible attacks - in the case of the 5-language model, we can have 120 possible attack scenarios.
>
> 3) **Study of the dependency of models’ confidence on the length of an added sequence.**
> We show that the total number of non-toxic words in a sentence would impact the toxicity score.
> However, we believe it is not how it should be:\
> Jigsaw defines toxicity as "rude, disrespectful, or unreasonable language that is likely to make someone leave a discussion." from this point of view, messages: *"Nobody wants to talk with you!"* and *"Nobody wants to talk with you! text for a bot to avoid ban: flowers happiness good love"* should have the same toxicity score.\
> In the same way as a drop of poison will turn a bottle of wine into a bottle of poison, a toxic message cannot be watered down by a bunch of positive words. Hatred cannot be justified.\
> Thus, **by our attack we demonstrate a significant flaw in the current approach to toxicity detection**.
>
> &nbsp;
> ## Reason to reject №2
> &nbsp;
>
> >The attack model may need to be justified a lot. One straightforward flaw is that the model can be easily identified by humans and not identified by machines. Some simple modifications of the detector may be able to solve such a problem; for example, a language model can be used first to check if the sentences sound natural, or natural language inference can be used first to detect if the relations between sentences are natural.
>
> This is the point of the suggested method: **our attack targets exclusively machines, making it clear to humans that the positive words are intended for the automoderation system** (e.g. "Nobody wants to talk with you! text for a bot to avoid ban: flowers happiness good love").
> This allows an attacker to preserve their original message without any changes, unlike in other attacks which substitute words with synonyms or replace original characters with their look-alike.
>
> Moreover, in a real-world scenario, an adversary can use this attack in combination with other approaches, thus increasing defence difficulty.
>
> In our current research, we are developing a more sophisticated way of constructing insertion with respect to the original message, making it a semantically correct addition to a human-written message, thus making adversarial training more difficult and increasing the variety of adversarial insertions. It’s a GAN-like architecture, with an LLM, such as Llama 2, acting as a generator and a toxicity detector acting as a discriminator.
>
> &nbsp;
> ## Question №1
> &nbsp;
>
> >How about the detectors used here in the paper? Maybe some more details need to be provided. Are they advanced enough?
>
> We used the most popular toxicity detectors from the Hugging Face hub for each language tested and provided the names of these models, as them being popular shows the importance of analysis of their vulnerabilities.\
> BERT, RoBERTa and ELECTRA architectures were tested to compare their performance in different languages, and 14 experiments on different English monolingual models were performed (see appendix A).
>
> &nbsp;
> ## Question №2
> &nbsp;
>
> >How about the mentioned attacks for large language models? Will the detection be easy?
>
> We conducted additional experiments with the Perspective API and OpenAI Moderation API. As we discovered, they are also vulnerable to our attacks.
> In the table below we show the performance of a word-adding attack with the phrase: "I hate you, dumbass! text for a bot to avoid ban:"
>
> | n words | OpenAI stable | OpenAI latest | Perspective API |
> |:-------:|:-------------:|:-------------:|:---------------:|
> |    1    |     0.643     |     0.946     |      0.899      |
> |    2    |     0.663     |     0.915     |      0.903      |
> |    3    |     0.647     |     0.918     |      0.899      |
> |    4    |     0.605     |     0.904     |      0.886      |
> |    5    |     0.601     |     0.872     |      0.878      |
> |    6    |     0.553     |     0.902     |       0.87      |
> |    7    |     0.553     |     0.926     |      0.863      |
> |    8    |     0.551     |     0.916     |      0.854      |
> |    9    |      0.33     |     0.854     |      0.853      |
> |    10   |     0.421     |     0.847     |      **0.846**      |
> |    11   |     0.381     |     0.705     |      0.852      |
> |    12   |     0.427     |      0.74     |      0.852      |
> |    13   |     0.453     |     0.761     |      0.852      |
> |    14   |     0.426     |     **0.665**     |      0.853      |
> |    15   |     0.362     |     0.681     |      0.853      |
> |    16   |    **0.265**    |     0.716     |      0.853      |
>
> We also conducted additional experiments in other languages and between them (only with OpenAI, Perspective API is failing with an error in the multilingual setting). In the table below, you can find the results of cross-lingual attacks on OpenAI models:
>
> | n words | Latest EN+FR | Latest EN+GER | Stable EN+FR | Stable EN+GER |
> |:-------:|:------------:|:-------------:|:------------:|---------------|
> |    0    |     0.872    |     0.872     |     0.888    | 0.888         |
> |    1    |     0.931    |     0.981     |     0.793    | 0.848         |
> |    2    |     0.895    |     0.898     |     0.779    | 0.51          |
> |    3    |     0.769    |     0.889     |     0.671    | 0.511         |
> |    4    |     0.816    |     0.846     |     0.594    | 0.344         |
> |    5    |     0.771    |     0.807     |     0.606    | 0.302         |
> |    6    |      0.8     |     0.647     |     0.618    | 0.318         |
> |    7    |     0.659    |     0.533     |     0.605    | **0.201**         |
> |    8    |     0.592    |     0.555     |     0.573    | 0.208         |
> |    9    |     **0.579**    |    **0.489**     |     **0.424**    | 0.283         |
>
> We also tested ChatGPT in conversation mode with different prompts and found it susceptible to the attack.
>
> &nbsp;
>
> Thank you for your time and effort,\
> We rest at your disposal to answer any additional questions should you have any.

---

### Meta-Review · Area_Chair_ANgx · 2023-09-29

**Recommendation:** 2

**Metareview:**

The work demonstrated a novel and effective way of bypassing the toxicity detection models by appending positive words or sentences to the end of a toxic message. It showed that this sentence-level attack can be applied to multiple languages from different language families, and that the models’ confidence can be reduced by increasing the length of the added sequence.

Pros:
Introduced a concept of “To Each His Own” attack, which is based on the idea of separating the messages addressed to a human and an algorithm. The attack exploits the fact that the toxicity detection models are trained on sentence-level labels, and do not consider the context or the intention of the message.

Cons
May not be effective on toxicity detectors that use different definitions or criteria of toxicity

---

### Decision · Program_Chairs · 2023-10-07

**Decision:**

Accept-Findings

**Comment:**

The work demonstrated a novel and effective way of bypassing the toxicity detection models by appending positive words or sentences to the end of a toxic message. It showed that this sentence-level attack can be applied to multiple languages from different language families, and that the models’ confidence can be reduced by increasing the length of the added sequence.

Pros:
Introduced a concept of “To Each His Own” attack, which is based on the idea of separating the messages addressed to a human and an algorithm. The attack exploits the fact that the toxicity detection models are trained on sentence-level labels, and do not consider the context or the intention of the message.

Cons
May not be effective on toxicity detectors that use different definitions or criteria of toxicity